# Autopsy for Medical Diagnostics: Finding the Cause of Sudden Unexpected Death through Investigation of the Cardiac Conduction System by Serial Sections

**DOI:** 10.3390/diagnostics13111919

**Published:** 2023-05-31

**Authors:** Giulia Ottaviani, Simone G. Ramos

**Affiliations:** 1Lino Rossi Research Center, Anatomic Pathology, Department of Biomedical, Surgical and Dental Sciences, Università degli Studi di Milano, 20122 Milan, Italy; 2Pathology and Legal Medicine, Ribeirão Preto Medical School, University of São Paulo, Ribeirão Preto 14040-900, Brazil; sgramos@fmrp.usp.br

**Keywords:** autopsy, post-mortem investigation, sudden unexpected death, sudden infant death syndrome, sudden intrauterine unexpected death, cardiac conduction system

## Abstract

Sudden unexpected death (SUD) is a fatal event that occurs in an apparently healthy subject in a way that such an abrupt outcome could have not been predicted. SUD—including sudden intrauterine unexplained death (SIUD), sudden neonatal unexpected death (SNUD), sudden infant death syndrome (SIDS), sudden unexpected death of the young (SUDY), and sudden unexpected death in the adult (SUDA)—occurs as the first manifestation of an unknown underlying disease or within a few hours of the presentation of a disease. SUD is a major unsolved, shocking form of death that occurs frequently and can happen at any time without warning. For each case of SUD, a review of clinical history data and performance of a complete autopsy, particularly focused on the study of the cardiac conduction system, were carried out according to the necropsy protocol devised by the Lino Rossi Research Center, Università degli Studi di Milano, Italy. Research cases collected and selected for this study were represented by 75 SUD victims that were subdivided into 15 SIUD, 15 SNUD, 15 SUDY, and 15 SUDA victims. After a routine autopsy and clinical history analysis, death remained unexplained, and hence a diagnosis of SUD was assigned to 75 subjects, which included 45 females (60%) and 30 (40%) males ranging in age from 27 gestational weeks to 76 years. Serial sections of the cardiac conduction system disclosed frequent congenital alterations of the cardiac conduction system in fetuses and infants. An age-related significant difference in distribution among the five age-related groups was detected for the following anomalies of the conduction system: central fibrous body (CFB) islands of conduction tissue, fetal dispersion, resorptive degeneration, Mahaim fiber, CFB cartilaginous meta-hyperplasia, His bundle septation, sino-atrial node (SAN) artery fibromuscular thickening, atrio-ventricular junction hypoplasia, intramural right bundle branch, and SAN hypoplasia. The results are useful for understanding the cause of death for all SUD cases that were unexpected and would have otherwise remained unexplained, so as to motivate medical examiners and pathologists to perform more in-depth studies.

## 1. Introduction

Sudden unexpected death (SUD) is the sudden death that occurs as the first manifestation of an unknown underlying disease, or within a few hours, and is unexpected by history. SUD is a common form of death, especially in the young, that represents a major public health problem with a devastating impact on the involved families and communities. The dynamic of this event poses little, if any, hope for survival. The emotional consequences among families of victims are devastating, with high social costs due to the loss of many potentially productive individuals [1] and for the psychological support programs needed for family members [2]. According to the most recent report from the American Heart Association [3], the incidence of out-of-hospital cardiac arrest in people of any age is 92.3 individuals per 100,000 population. About 50% of sudden deaths have no specific findings in the autopsy and the mechanism of death is classified as arrhythmic in nature.

SUD can hit human individuals of any age, gender, ethnicity, or nationality. SUD includes sudden intrauterine unexpected death (SIUD), sudden neonatal unexpected death (SNUD), sudden infant death syndrome (SIDS), sudden unexpected death of the young (SUDY), and sudden unexpected death in the adult (SUDA). The broad spectrum of SUD may have common risk factors affecting individuals of all ages, such as obstructive sleep apnea syndrome (OSAS) [4].

SIDS, or crib death, is the sudden unexpected death of an infant, with the onset of the fatal episode apparently occurring during sleep, which remains unexplained after a thorough investigation, including the performance of a complete autopsy and review of the clinical history [5]. SIDS is the most frequent death-causing syndrome in the first year of life, with a birth prevalence of 0.35% [6]. Since its definition was introduced by the National Institute of Child Health and Human Development (NICHD) panel in 1991 [7], three decades of SIDS research have revealed only parts of its anatomopathological underlying features that interplay when determining SIDS.

Sudden intrauterine unexpected death (SIUD), or unexpected stillbirth, is late fetal death before the complete expulsion or removal of the fetus from the mother at ≥25 weeks of gestation, which is unexpected by history and is unexplained after a review of the maternal clinical history and the performance of a general autopsy of the fetus, including examination of the placental disk, umbilical cord, and membranes [8].

The causes of SUD are largely unknown due to a lack of specialized post-mortem and clinical studies. The risk factors for SUD are largely unknown as the body of clinicopathological and epidemiological research to explain SUD is scarce and fragmentary. We hypothesize that there are common risk factors for SUD affecting individuals of all ages, starting from a victim’s intrauterine existence. Except for our few previous reports [9,10,11,12], there are no studies on the cardiac conduction system in all age groups of subjects dying suddenly and unexpectedly.

This study aims to evaluate, in a more complete way, a series of cases of SUD by natural causes, according to the necropsy protocol devised by the Lino Rossi Research Center, Università degli Studi di Milano [13,14], in order to determine the presence and significance of the post-mortem findings in the cardiac conduction system, distributed by age groups.

## 2. Materials and Methods

### 2.1. Selection and Classification of Cases

The Lino Rossi Research Center of the Università degli Studi di Milano, Milan, Italy, is a specialist referral center for cases of suspected sudden unexpected death (SUD) across Italy, with a database of over 1000 cases. Close relatives or parents of deceased subjects provided written informed consent to autopsy examination, according to protocols approved by the institutional review board (IRB) of the Lino Rossi Research Center, with the Department of Biomedical, Surgical and Dental Sciences, Università degli Studi di Milano, Milan, Italy providing funding support PSR 2020, linea 2 on 20 November 2020.

All cases of unexpected infant and perinatal death referred to the Lino Rossi Center underwent anatomoclinical investigations, according to Italian law No. 31 of 2/02/06 “Regulations for diagnostic post-mortem investigation in victims of sudden infant death syndrome (SIDS) and of unexpected fetal death” [15]. Confidentiality and privacy in personal data collection and processing were ensured following the European and U.S. legislation on these matters.

In the referred cases, natural death presumably due to subtle cardiovascular or brainstem lesions was suspected because no lesions found in the general autopsy could explain death, even after the exclusion of non-natural causes of death and after the toxicological examinations were negative for drug or alcohol abuse. For the purposes of this work, the study population consists of 75 consecutive autopsy cases of SUD submitted to the Lino Rossi Research Center of the Università degli Studi di Milano after the performance of a general autopsy was not able to establish the cause of death. The lack of the cause of death led us to seek a more in-depth anatomopathological investigation.

Based on the age at death, SUD cases were sub-classified post-mortem in five categories, as follows:(1)Sudden intrauterine unexpected death (SIUD) is when a fetus, from the 25th gestational week to term, died suddenly and unexpectedly before the complete expulsion or extraction of the fetus from the mother, resulting in a stillbirth for which there was no explanation after review of maternal clinical history and the performance of a general autopsy of the fetus, including examination of the fetal adnexa, i.e., placental disk, umbilical cord, and membranes [8,11,13].(2)Sudden neonatal unexpected death (SNUD) is when a newborn, aged from birth to one month, died suddenly, unexpectedly by history and unexplainably after a thorough case investigation, including the performance of a general autopsy, an examination of the death scene, and a review of clinical history [8,13,16].(3)Sudden infant death syndrome (SIDS) is when death in an infant, aged one month to one year, was sudden, unexpected by history, and unexplained after a thorough case investigation, including the performance of a general autopsy, an examination of the death scene, and a review of clinical history [5,8].(4)Sudden unexpected death in the young (SUDY) is when death occurred in an individual aged 1–35 years, suddenly and unexpectedly in an apparently healthy subject in a way that such an abrupt outcome could have not been predicted, unexplained after a review of the scene of death and clinical history, and the performance of a general autopsy.(5)Sudden unexpected death in the adult (SUDA) is when death occurred in an individual older than 35 years, suddenly and unexpectedly in an apparently healthy subject in a way that such an abrupt outcome could have not been predicted, unexplained after a review of the scene of death and clinical history, and the performance of a general autopsy.

A case was classified as sudden unexpected death (SUD)—including SIUD, SNUD, SIDS, SUDY, and SUDA—when death occurred suddenly since the beginning of symptoms and was unexpectedly by history.

A SIUD or SIDS case was classified as gray zone or borderline when it was unexpected by history and unexplained after a review of clinical history and performance of a general autopsy, which disclosed another event that acted as a triggering phenomenon, which itself was not enough to cause death in vulnerable individuals [17,18].

A total of 75 consecutive autopsy cases referred to at the Lino Rossi Center were retrospectively enrolled into the study if they met the pathological criteria to be divided into the two following groups for comparison: (1) the SIUD group: 15 cases; (2) the SNUD group: 15 cases; (3) the SIDS group: 15 cases; (4) the SUDY group: 15 cases; (5) and the SUDA group: 15 cases.

### 2.2. Necropsy Investigational Protocol

For each study case, a full review of clinical history data and performance of a complete autopsy study were carried out according to the necropsy protocols devised by the Lino Rossi Research Center, Anatomic Pathology, Università degli Studi di Milano, Milan, Italy [13,14,18]. In particular, in all cases, an in-depth histopathological examination of the cardiac conduction system and the brainstem on serial sections was performed, with the principal aim of detecting fine alterations in structures controlling vital functions.

The clinical information of the SUD victims was provided by the referring pathologists or coroners.

A complete autopsy examination was carried out, including a systemic gross and microscopic evaluation of the body. In fetuses, the placental disk, umbilical cord, and membranes were examined. All organs were fixed in 10% phosphate-buffered formalin, processed, and paraffin-embedded. The cardiac conduction system was the particular focus of this study.

Each heart was regularly examined for pathological changes in the atria, septa, ventricles, pericardium, endocardium, and coronary arteries. The origin of the coronary arteries was carefully inspected. Multiple samples of the major coronary arteries (left main, left anterior descending, left circumflex, right main, right posterior descending, right marginal) were collected and examined. Samples of the myocardium and coronary arteries were stained with Hematoxylin–Eosin (HE) and Trichromic Heidenhain (Azan).

For the morphological study of the conduction system, two specimens of the heart were obtained for paraffin embedding. The first specimen contained the sino-atrial node (SAN), its atrial approaches, the *crista terminalis*, and the SAN gangliar plexus. The main visual reference for the removal was centered upon the *sulcus-crista terminalis*. Two longitudinal cuts were driven, parallel to the sulcus-crista line, through the atrial wall with a medial prolongation on the right side to encompass the anterior aspect of the inlet of the superior vena cava. On the left side, the cava–cava bridge was sectioned medially, prolonging the cut on the superior vena cava wall. The second specimen contained the atrioventricular (AV) junction with its atrial approaches. The pinpoints of the excision were, on the right side, the outlet of the coronary sinus and the *pars membranacea septi*. The following cuts were driven: an inferior, longitudinal incision through the posterior part of the septum, across the AV annulus fibrous and up the superior margin of the coronary sinus ostium; an anterior longitudinal incision parallel to the former, through the superior part of the septum, extending to the aortic valvular ring; and two cuts perpendicular to the previous two cuts. Both samples were routinely fixed in 10% buffered formalin and paraffin-embedded. The sections were cut at intervals of 20–40 μm (levels). For each level, five 8 μm sections were retained, mounted, and alternately stained with HE and Azan. All intervening sections were kept and stained as deemed necessary [11,18].

### 2.3. Statistical Analysis

Quantitative data were expressed as means ± SD. The significance of differences between group parameters was evaluated by Student’s *t*-test, a chi-square test, or Fisher’s test. In the case of a skewed distribution, a nonparametric Whitney rank-sum test was used. A one-way ANOVA test for continuous variables was used for quantifying and partitioning variance between groups. When the results suggested differences, pairwise differences were assessed by Bonferroni’s post hoc test. The relationship between variables was analyzed by a Pearson correlation test. Statistics were compiled using SigmaStat^®^ (version 4, Systat Software Inc., Chicago, IL, USA) and plotted using SigmaPlot^®^ (version 14, Systat Software Inc., Chicago, IL, USA) statistical software. The selected level of significance was *p* < 0.05 and was two-tailed.

## 3. Results

In accordance with the aim of this study, an in-depth necropsy examination of victims of sudden unexpected death (SUD) of natural causes was carried out by grouping the victims according to their age at death.

### 3.1. Demographic and Clinical Data

Of a series of over 1000 cases submitted to the Lino Rossi Research Center of the Università degli Studi di Milano, Milan, Italy, for specialized anatomopathological post-mortem analyses collected from January 1987 to date, 75 consecutive victims of sudden unexpected death (SUD) divided by age were selected for this study. The demographic and clinical data are shown in Table 1.

The 75 SUD victims enrolled in this study, referred to the Lino Rossi Center, Università degli Studi di Milano for more in-depth investigations, were subdivided into the following five age-related groups:(1)Fifteen victims of sudden intrauterine unexpected death (SIUD) aged 25th gestational week (gw) to birth;(2)Fifteen victims of sudden neonatal unexpected death (SNUD) aged from birth to one postnatal month;(3)Fifteen victims of sudden infant death syndrome (SIDS) aged from one postnatal month to one postnatal year;(4)Fifteen victims of sudden unexpected death in the young (SUDY) aged 1–35 years (yrs);(5)Fifteen victims of sudden unexpected death in the adult (SUDA) older than 35 years, as shown in Table 1.

Accordingly, in this study, in the SUD cases analyzed, a diagnosis of SIUD was established in fifteen fetuses, nine (60%) males and six (40%) females in the age range of 27–41 gws (mean age ± SD, 37.39 + 3.69 gws). The fifteen SNUD newborns comprised ten (66.67%) males and five (33.33%) females in the age range of 1 h–28 days after birth (mean age ± SD, 7.08 + 9.92 days). The fifteen SIDS infants comprised seven (46.67%) males and eight (53.33%) females in the age range of 31–225 postnatal days (mean age ± SD, 105.67 + 59.84 days). The fifteen SUDY young subjects comprised ten (66.67%) males and five (33.33%) females in the age range of 15 months–33 yrs (mean age ± SD, 22.92 + 10.33 yrs). The fifteen SUDA adults comprised nine (60%) males and six (40%) females in the age range of 36–76 yrs (mean age ± SD, 49.69 + 11.72 yrs) (Table 1).

Among the 75 SUD cases, ranging in age from 27 gws–76 yrs, the 45 (60%) males were statistically more frequent than the 30 (40%) females (Figure 1).

Two cases of SIUD were classified as a SIUD gray zone, respectively, due to concomitant amniotic fluid aspiration and chorioamnionitis. One case of SIDS was classified as a SIDS gray zone due to a concomitant diagnosis of malaria. Three cases were classified as an SNUD gray zone due to coexistent multiple cerebral ventricular and periventricular hemorrhages in one case, and pneumonia in two cases.

The investigation of the SUDY cases disclosed the anatomopathological diagnosis of dilated cardiomyopathy (DCMP) in one case (6.67%), hepatic insufficiency in two cases (13.33%), myocarditis in three cases (20%), coronary anomalous origin in one case (6.67%), myocardial infarction in two cases, and arrhythmogenic right ventricular cardiomyopathy (ARVC) in five cases (33.33%). The clinicopathological findings of the SUDA cases disclosed DCMP in one case (6.67%), myocarditis in two cases (13.33%), lymphoma metastases in two cases (13.33%), myocardial infarction in two cases (13.33%), and ARVC in five cases (33.33%). The findings of the conducting tissue of the ARVC cases, as hypoplasia in the conduction system due to fibro-fatty tissue infiltration, which now belong to the groups of SUDY and SUDA, were previously reported [12].

### 3.2. Cardiac Conduction System Findings

The 75 SUD cases, in which the CCS has been fully analyzed, consisted of 15 cases of SIUD, 15 cases of SNUD, 15 cases of SIDS, 15 cases of SUDY, and 15 cases of SUDA.

In the 75 SUD cases, the following anomalies of the CCS have been detected in the following percentages of cases: islands of conduction tissue in the central fibrous body (CFB) in 54.67%; fetal dispersion in 46.67%; resorptive degeneration in 45.33%; Mahaim fiber in 45.33%; CFB cartilaginous meta-hyperplasia in 30.67%; septated bifurcation in 24%; septated His bundle (HB) in 24%; fibromuscular thickening of the sino-atrial node (SAN) artery in 14.67%; fibromuscular thickening of the atrio-ventricular node (AVN) artery in 13.33%; AVN duplicity in 10.67%; intramural bifurcation in 10.67%; hemorrhage and infarct-like lesions in the atrio-ventricular junction (AVJ) in 10.67%; AVJ hypoplasia in 9.33%; intramural right bundle branch (RBB) in 9.33%; CBF hypoplasia in 6,67%; SAN hypoplasia in 5.33%; septated AVN in 4%; AVN tongue in 2.67%; intramural left bundle branch (LBB) in 2.67%; intramural His bundle (HB) in 2.67%; HB duplicity in 2.67%; and SAN hemorrhage and infarct-like lesions in 1.3% (Table 1).

An age-related statistically significant difference in distribution among the five age-related groups was detected for the following anomalies of the conduction system: CFB islands of conduction tissue, fetal dispersion, resorptive degeneration, Mahaim fiber, CFB cartilaginous meta-hyperplasia, HB septation, SAN artery fibromuscular thickening, AVJ hypoplasia, intramural RBB, and SAN hypoplasia (Table 1).

Although some lesions were detected only in some groups, the AVJ hypoplasia was detected only in SUDY and SUDA, and the AVN tongue was detected only in the SUDY group; the difference among groups was not statistically significant and was not age-related (Table 1).

We observed islands of conduction in the CFB system more frequently in SNUD (86.67%) and less frequently in SUDA (26.67%) (Table 1). In such cases, we observed islands of AVJ inside the CFB (Figure 2) as a form of persistent fetal dispersion, in part undergoing the process of resorptive degeneration (Figure 2B), which was the process of molding and reabsorption of the AVJ from the embryonic toward the adult shape [9,10,11,12].

We observed areas of fetal dispersions in the forms of AVJ undergoing the process of resorptive degeneration (Figure 3) in the vast majority (80%) of both SIUD and SNUD cases. Separated areas of resorptive degeneration were mostly seen in SIDS (93.33%) cases. Mahaim fibers, as accessory pathways connecting the AVJ directly with the myocardium of the interventricular septum (Figure 3), were mostly detected in SIDS (86.67%) cases. An AVN tongue was detected in 2.67% of SUD cases (Table 1).

We observed cartilaginous meta-hyperplasia of the CFB, potentially compressing the AVJ (Figure 4) in 36.67% of SUD cases absent in the SUDA group, with an age-related distribution (Table 1).

HB septation, characterized by fibrous tissue infiltration from the CFB into the HB (Figure 5A), was mostly present in the SIDS group (46.67%) with an age-related distribution. Septation of the bifurcation (BF), or anomalous bifurcation (Figure 5B), was present in all age groups without significant age-related differences. Septated AVN was detected in only 4% of SUD cases (Table 1).

Fibromuscular thickening of the sino-atrial node (SAN) artery, as fibromuscular dysplasia (Figure 6), has been detected in 14.67% of SUD cases, which is correlated with age increase more frequently in the SUDA group (33%) (Table 1).

Fibromuscular thickening of the AVN artery, as fibromuscular dysplasia, has been detected in 13.33% of SUD cases. Although it was absent in the SIUD group and was more frequent in the SUDA group, the age distribution was not statistically significant (Table 1). AVJ duplicity, as AVN and HB duplicity and as split AVJ, was observed in 10.67% and 2.67% of SUD cases, respectively, without significant age differences (Table 1). Intramural AVJ, as intramural BIF, right bundle branch (RBB), and left bundle branch (LBB), was observed in the decrescent prevalence of 10.67%, 8%, and 2.67% of SUD cases, respectively; also, the intramural HB was observed in 2.67% of cases, without age-related differences (Table 1). The AVJ and the SAN hemorrhage and infarct-like lesions, due to cardiac massage, were detected in 10.67% and 1.3% of SUD cases; they were absent in the SIUD group, but differences were not significant (Table 1). We observed hypoplasia of the AVJ, CFB, and SAN (Figure 6) in 9.33%, 6.67%, and 5.33% of SUD cases, respectively, with age-related differences, except for CFB hypoplasia (Table 1).

## 4. Discussion

Sudden unexpected death (SUD) is a fatal event that occurs in an apparently healthy subject in a way that such an abrupt outcome could have not been predicted. The emotional consequences among families are devastating and have a high social cost, considering the post-traumatic stress disorders of family members [1,19].

Once an unnatural cause of death is ruled out by the medical examiner, little attention, if any, is given to identifying the pathogenesis and risk factors of SUD. There is a broad spectrum of SUD with possible common risk factors affecting individuals of all ages, starting from intrauterine life. The anatomoclinical substrates of SUD are largely unknown [20]. The incidence of SUD peaks in infancy and, in adults, it increases exponentially with age, surpassing the risk for infants by 35–39 years of age [21]. OSAS is the most common form of sleep-disordered breathing that affects approximately 10% of adults and 5% of children [21]. Anastasakis et al. [22] reported that sudden unexpected death of the young (SUDY) had a cardiovascular origin in 65% of cases and remains unexplained in 18% of cases after autopsy investigations.

Unexpected stillbirth, or SIUD, has a 6-8 fold greater incidence than SIDS that has not significantly declined in the last three decades, despite modern advances in maternal-infant care [23].

Despite the valuable anatomopathological research over the last 30 years to explain SIDS and SIUD [9,10,24,25,26], overall, the body of literature on the findings in the heart is still scarce and fragmentary.

Among the congenital anomalies detected in SIDS and SIUD, the present abnormalities of the cardiac conduction system, including resorptive degeneration, Mahaim fibers, and cartilaginous meta-hyperplasia (Figure 2, Figure 3 and Figure 4), have been reported along with several neuropathology anomalies of the autonomic nervous system structures, mainly the hypoplasia of the arcuate nucleus, the hypoplasia of the respiratory reticular formation, and the parabrachial/Kölliker–Fuse complex, which are significantly related to maternal cigarette smoking in over 50% of cases [27,28,29,30,31,32].

Current guidelines for SIDS and other sudden arrhythmic death syndromes state the potential usefulness of a molecular autopsy, as cardiac channelopathies are regarded as a major genetic contributor to these sudden and otherwise unexplained deaths [20]. Massively parallel DNA sequencing systems provide a sequence of great numbers of different DNA strands at once. These technologies are revolutionizing our understanding of medical genetics, accelerating health improvement projects, and laying the groundwork to fully understand personalized medicine in the near future. Whole-exome sequencing (WES) is the application of next-generation technology to determine the variations of all coding regions, or exons, of known genes. Sequencing of the complete coding regions (exomes) has the potential to uncover the causes of SIDS-SIUDS. There is very little knowledge about the WES application in the early diagnosis of potential SIDS-SIUD victims and healthy carriers, and their utility in clinical screening is also unknown.

A post-mortem examination is of great importance in every case of SUD sine *materia*, with the examination of the cardiac conduction system and brainstem on serial sections, when a general autopsy fails to find the cause of death after the exclusion of violent causes of death and the absence of any explained cause of death by a general autopsy.

### 4.1. Subjects’ Characteristics

In this study, among the 75 SUD subjects grouped by age from consecutive autopsy cases referred to at the Lino Rossi Center, Università degli Studi di Milano for more in-depth investigations, 15 were victims of sudden intrauterine unexpected death (SIUD), aged 25th gw to birth, 15 were victims of sudden neonatal unexpected death (SNUD), aged from birth to one postnatal month, 15 were victims of sudden infant death syndrome (SIDS), aged from one postnatal month to one postnatal year, and 15 were victims of sudden unexpected death in the young (SUDY), aged 1–35 years and sudden unexpected death in the adult (SUDA), older than 35 years. (Table 1).

SUD of all ages was more frequent in males (60%) than females (40%) (Figure 1). Regarding SIUD, in the present study, we are reporting a higher incidence of males (60%) than females (40%) (Figure 1, Table 1). In a previous report by Ottaviani et al. [11], another sample of 15 SIUD cases from our series presented a higher incidence in females (60%).

In the SNUD group, we found 66.67% males versus 33.33% females (Figure 1, Table 1). More epidemiological studies in larger series are needed to define the gender predominance in sudden death occurring in the perinatal age, including both SIUD and SNUD.

Regarding SIDS, in the present study, the percentage of females (53.33%) has been surprisingly higher than the percentage of males (46.67%) (Figure 1, Table 1). In a previous work, Matturri et al. [33] reported from our series of 102 cases that males were 62.74% versus 37.26% for females. The male sex has been considered to be a risk factor for SIDS along with prone position, death during sleep, co-sleeping, and preterm birth, among others [34,35].

In this study, the findings of SIUD, SNUD, and SIDS gray zones are consistent with the proposed definition by Ottaviani [8,17] of the gray zone or borderline cases. We suggest that the concept of SIDS-gray zone, already described in the SIDS-SIUD complex [8], be applied to SUD cases of all ages, whenever an in-depth serial sections study of the conducting tissue would discover anomalies able to complete the puzzle of the cause of death in subjects with other pathological findings, such as ARVC in adults.

SUDY disclosed one case of DCMP (6.67%) and five cases of ARVC (33.33%). The percentages slightly differ from the reported prevalence of cardiomyopathy types in other series [36], as they depend on the need for a diagnosis and the complimentary experience that pathologists were seeking in our institution. Thiene et al. [37] reported 20% of sudden deaths in young athletes as “sine-materia”. Such cases could be explained due to anomalies of the cardiac conduction system, as we are now reporting them for this age group.

In the literature, there is not a clear consensus definition of SUD. In our study, SUD encompassed all types of sudden unexpected death by natural causes of all ages, from pre-natal life to old age, including SIUD, SNUD, SIDS, SUDY, and SUDA. SIUD, SNUD, and SIDS have been grouped in the so-called SIDS-SIUD complex [8]. We suggest that they could all be part of sudden arrhythmic death syndrome (SADS), as dysfunctions of the cardiac conduction and autonomic nervous systems are known to contribute to the pathogenesis of SADS [20].

### 4.2. Anatomopathological Findings in the Cardiac Conduction System

Research cases collected and selected for this study at the Lino Rossi research at the Università degli Studi di Milano, Italy were represented by 75 SUD victims that were subdivided into 15 SIUD, 15 SNUD, 15 SUDY, and 15 SUDA victims. After the routine autopsy and clinical history analysis, death was unexpected, and hence, a diagnosis of “SUD” was made for the 75 subjects, consisting of 45 females (60%) and 30 (40%) males ranging in age from 27 gws to 76 yrs. Serial sections of the cardiac conduction system were carried out to provide data on both the topography and pathology of the specialized AV pathways.

As shown in Table 1, an age-related significant difference in distribution among the five age-related groups was detected for several anomalies of the conduction system, including CFB islands of conduction tissue (Figure 2), fetal dispersion, resorptive degeneration, Mahaim fiber (Figure 3), CFB cartilaginous meta-hyperplasia (Figure 4), HB septation (Figure 5), SAN artery fibromuscular thickening, and SAN hypoplasia (Figure 6) (Table 1). The fact that the same anomalies are found in all groups suggests their congenital nature.

The anomalies found in the conducting system could be the morphological bases for the development of cardiac arrhythmias. Islands of the conduction system in the CFB could act as potential arrhythmogenic ectopic foci in all age groups. In 1968, James [38] suggested the idea that the resorptive degeneration of the AVJ was the process by which the conducting system of infants goes through partial reabsorption toward its adult shape. According to Matturri et al. [39], apoptosis takes part in this reabsorption process and leaves accessory pathways or ectopic foci of the conduction system if it is defective. On the other hand, according to James [40], an exaggerated resorptive degeneration can cause hypoplasia of the AVJ. In this study, resorptive degeneration has been detected in all age groups, demonstrating its role in the sudden unexpected death of all ages.

In this study, some anomalies of CCS lesions were detected mostly in fetuses and infants rather than adults, such as cartilaginous meta-hyperplasia of the CFB (Table 1) (Figure 4) and AVN duplicity. Some anomalies were not detected in the SIUD, SNUD, and SIDS groups, such as AVJ and SAN hypoplasia (Table 1) (Figure 6), which have been reported to be related to metastatic infiltration or ARVC involvement in the adult groups [12,41,42]. AVN tongue was not detected in fetuses and infants (Table 1), but it was reported to be present in 13% of the SIUD cases in a previous study [11]. More research is needed, as there is a lack of studies on the cardiac conduction system in all age groups of subjects dying suddenly and unexpectedly.

Islands of AVJ inside the CFB (Figure 2) are also known as a form of persistent fetal dispersion [9,11] but they differ from fetal dispersion (Figure 3), which is itself a normal pattern of the AVJ distribution in fetuses.

Intramural HB, bifurcation, LBB, or RBB (Table 1) can be grouped into intramural AVJ and are morphological bases for cardiac arrhythmias [18] based on the different locations of the normal conductive pathways.

Hemorrhage and infarct-like lesions of the AVJ or SAN (Table 1) have been considered to be a consequence of cardiac massages [43,44]. In this study, they were not detected in the SIUD group, given that usually a cardiac massage is not performed in fetuses unless death occurs during birth.

Fibromuscular thickening of the SAN or AVN artery (Figure 6), as fibromuscular dysplasia, is itself not of a clear atherosclerotic nature [25] and has been detected in 14.67% of SUD cases and is correlated with age increase more frequently in the SUDA group (33%) (Table 1). In a previous study by Matturri et al. [25], fibromuscular thickening has been detected in infants and was suggested to be an early stage of atherosclerotic lesions that was correlated with parents’ smoking.

Our cases of SUD of all ages were unexpected by history, and a cause of death was not identified after a routine general autopsy but was rather explained by a detailed post-mortem investigation of the cardiac conduction system.

The study’s methods of investigation of the conduction system through serial sections will have a potential application for any form of sudden death occurring at any age. Potential mechanisms underlying these frequent morphological bases for sudden death are often overlooked. This study’s methods of investigation of the conduction system should have a potential application for any form of sudden death occurring at any age, in combination with other methods of post-mortem examinations that provide a strong complementary tool to the conventional autopsy. This is the case, for instance, of post-mortem cardiac magnetic resonance imaging, which offers a better understanding of the cardiovascular diseases responsible for sudden cardiac death [45]. However, the applied method of cardiac sampling to analyze the cardiac conduction system on serial sections is expensive, time-consuming, and requires trained personnel, both cardiovascular pathologists and laboratory histotechnicians.

The results are useful for understanding the cause of death for those unexpected cases that would have otherwise remained unexplained, so as to motivate medical examiners and pathologists to perform more in-depth studies.

More research is needed to explore congenital alterations involved in SUD pathogenesis and progression, such as the hypodevelopment of the cardiac conduction system and brainstem nuclei; to explore the risk factors of any clinical conditions, such as obstructive sleep apnea syndrome (OSAS) and explore the relationship between them and SUD; and to increase the knowledge on SUD in order to ultimately prevent this thorny problem and mitigate social concern over the issue.

## 5. Conclusions

SUD—including sudden intrauterine unexpected death (SIUD), sudden neonatal unexpected death (SNUD), sudden infant death syndrome (SIDS), sudden unexpected death of the young (SUDY), and sudden unexpected death in the adult (SUDA)—occurs as the first manifestation of an unknown underlying disease or within a few hours of the presentation of a disease. SUD is a major unsolved and shocking form of death that occurs frequently and can happen at any time without warning.

This study analyzed the pathological findings of the SUD of all ages, discussing the potential mechanisms underlying these frequent but overlooked and not always considered to be correlated forms of death. A post-mortem examination is of great importance in every case of SCD *sine materia*, with the examination of the cardiac conduction system on serial sections, when a general autopsy fails to find the cause of death.

More research is needed, as there is a lack of studies on the cardiac conduction system in all age groups of subjects dying suddenly and unexpectedly.

## Figures and Tables

**Figure 1 diagnostics-13-01919-f001:**
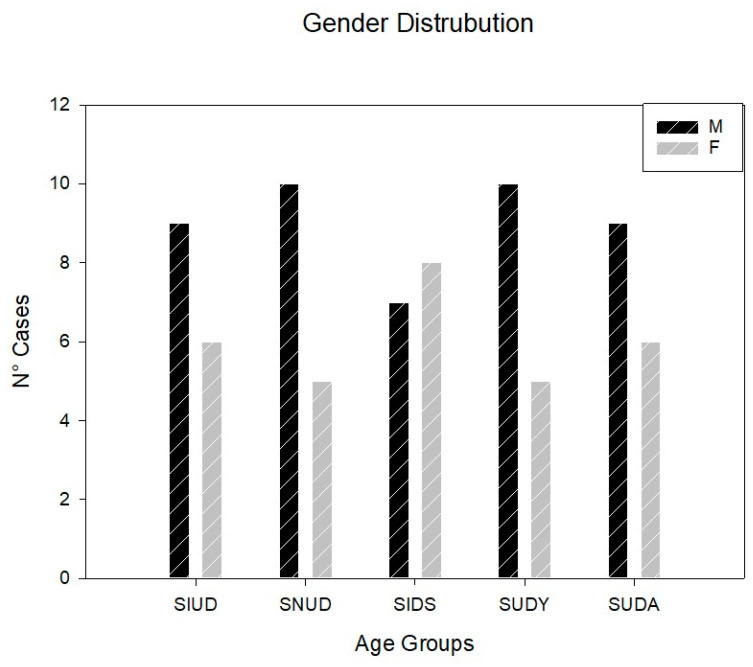
Gender and frequency distribution. This graph shows the gender and the frequency distribution of the 75 cases of sudden unexpected death (SUD) based on five age-related forms of death. SUD was significantly more frequent in males (M) (45/75) than females (F) (30/75), as the gender distribution among the five diagnostic subgroups between males (60%) and females (40%) was statistically significant (*p* < 0.05). SIUD = sudden intrauterine unexpected death; SNUD = sudden neonatal unexpected death; SIDS = sudden infant death syndrome; SUDY = sudden unexpected death in the young; SUDA = unexpected death in the adult.

**Figure 2 diagnostics-13-01919-f002:**
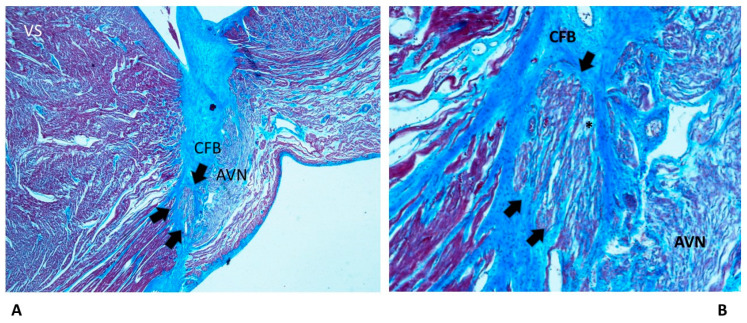
A histopathological slide of the atrioventricular junction of a 28-day-old baby boy victim of sudden neonatal unexpected death (SNUD). Arrows point to an island of conduction system embedded into the central fibrous body (CFB), separated from the atrioventricular node (AVN), in part undergoing the process of resorptive degeneration (*).VS = ventricular septum. Trichromic Heidenhain; original magnifications: (**A**) 20× and (**B**) 100×.

**Figure 3 diagnostics-13-01919-f003:**
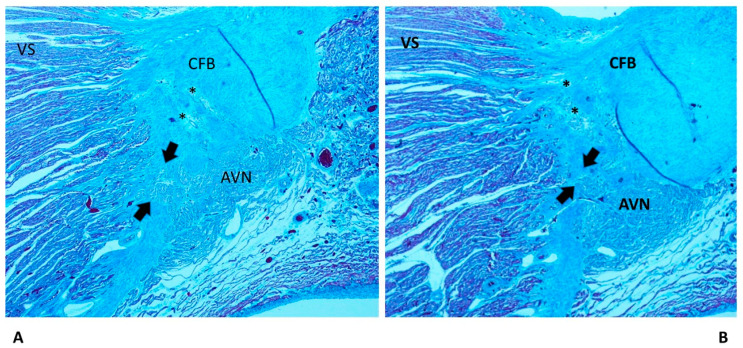
Two serial sections of the atrioventricular junction (AVJ) showing a Mahaim fiber (arrows) of a 39^+6^-week-gestation female fetus, a victim of sudden intrauterine unexpected death (SIUD). In the central fibrous body (CFB), the pattern of fetal dispersion, separated from the atrioventricular node (AVN), in part undergoing the process of resorptive degeneration (*), is shown. VS = ventricular septum. Trichromic Heidenhain; original magnification: (**A**,**B**) 20×.

**Figure 4 diagnostics-13-01919-f004:**
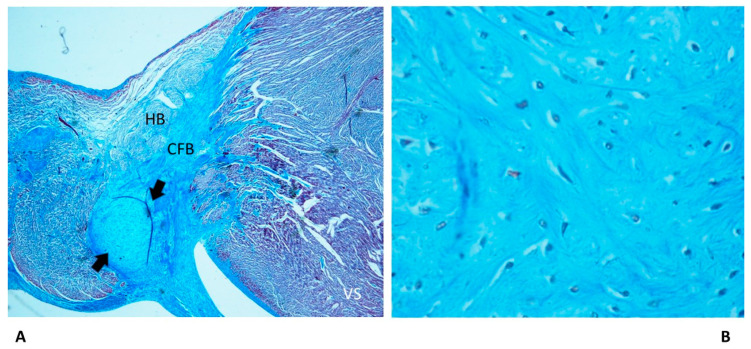
A histopathological slide of the atrioventricular junction of a 53-day-old baby girl victim of sudden infant death syndrome (SIDS). Arrows point to the cartilaginous meta-hyperplasia of the central fibrous body (CFB), potentially compressing the His bundle (HB). VS = ventricular septum. Trichromic Heidenhain; original magnifications: (**A**) 20× and (**B**) 100×.

**Figure 5 diagnostics-13-01919-f005:**
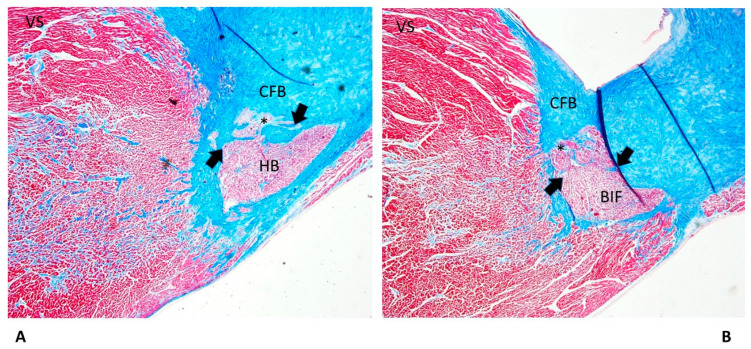
Two serial sections of the atrioventricular junction (AVJ), of a 13-year-old boy, a victim of sudden unexpected death in the young sudden intrauterine unexpected death (SUDY) group. (**A**) Arrows point to the fibrous tissue of the central fibrous body (CFB) as the pattern of septated His bundle (HB), which is in part undergoing the process of resorptive degeneration (*). (**B**) Arrows point to septation of the bifurcation (BIF). VS = ventricular septum. Trichromic Heidenhain; original magnification: (**A**,**B**) 20×.

**Figure 6 diagnostics-13-01919-f006:**
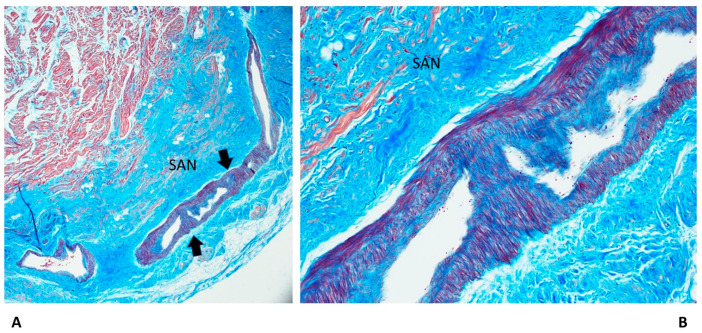
A histopathological slide of the sino-atrial node (SAN) of a 50-year-old man victim of sudden unexpected death in the adult (SUDA). Arrows point to fibromuscular thickening of the SAN artery as fibromuscular dysplasia. The SAN presents hypoplasia. Trichromic Heidenhain; original magnifications: (**A**) 20× and (**B**) 100×.

**Table 1 diagnostics-13-01919-t001:** Demographic data and histopathological findings of the 75 analyzed cases, divided by sudden intrauterine unexpected death (SIUD), sudden neonatal unexpected death (SNUD), sudden infant death syndrome (SIDS), sudden unexpected death in the young (SUDY), and sudden unexpected death in the adult (SUDA) groups.

	SIUD (*N* = 15)	SNUD (*N* = 15)	SIDS (*N* = 15)	SUDY (*N* = 15)	SUDA (*N* = 15)	Total (*N* = 75)
No. of cases	15	15	15	15	15	75
Gender (M/F)	9/6	10/5	7/8	10/5	9/6	45/30
Age range; mean ± SD	27–41 gw; 37.39 ± 3.69 gw	1 hr–28 dy; 7.08 ± 9.92 dy	31–225 dy; 105.67 ± 59.84 dy	14 ms–33 yr; 22.92 ± 10.33 yr	36–76 yr; 49.69 ± 11.72 yr	27 gw–76 yr
CFB: Islands of conduction tissue	10 (66.67%)	13 (86.67%)	8 (53.33%)	6 (40%)	4 (26.67%)	41 (54.67%) *
Fetal dispersion	12 (80%)	12 (80%)	9 (60%)	4 (26.67%)	3 (20%)	35 (46.67%) *
Resorptive degeneration	10 (66.67%)	14 (93.33%)	7 (46.67%)	2 (13.33%)	2 (13.33%)	34 (45.33%) *
Mahaim fiber	5 (33.33%)	10 (66.67%)	13 (86.67%)	3 (20%)	3 (20%)	34 (45.33%) *
CFB cartilaginous meta-hyperplasia	5 (33.33%)	4 (26.67%)	12 (80%)	2 (13.33%)	—	23 (30.67%) *
BIF: Septated	5 (33.33%)	6 (40%)	1 (6.67%)	2 (13.33%)	4 (26.67%)	18 (24%)
HB septated	1 (6.67%)	2 (13.33%)	7 (46.67%)	1 (6.67%)	1 (6.67%)	12 (16%) *
Fibromuscular thickening SAN artery	—	—	3 (20%)	3 (20%)	5 (33%)	11 (14.67%) *
Fibromuscular thickening AVN artery	—	1 (6.67%)	4 (26.67%)	1 (6.67%)	4 (26.67%)	10 (13.33%)
AVN duplicity	1 (6.67%)	4 (26.67%)	2 (13.33%)	1 (6.67%)	—	8 (10.67%)
BIF intramural	2 (13.33%)	3 (20%)	1 (6.67%)	1 (6.67%)	1 (6.67%)	8 (10.67%)
AVJ: Hemorrhage and infarct-like lesions	—	3 (20%)	1 (6.67%)	3 (20%)	2 (13.33%)	8 (10.67%)
AVJ hypoplasia	—	—	—	1 (6.67%)	6 (40%)	7 (9.33%) *
RBB: Intramural	4 (26.67%)	1 (6.67%)	—	—	1 (6.67%)	6 (8%) *
CFB hypoplasia	2 (13.33%)	—	—	2 (13.33%)	1 (6.67%)	5 (6.67%)
SAN hypoplasia	—	—	—	4 (26.67%)	2 (13.33%)	4 (5.33%) *
AVN septated	—	1 (6.67%)	1 (6.67%)	1 (6.67%)	—	3 (4%)
AVN tongue	—	—	—	2 (13.33%)	—	2 (2.67%)
LBB: Intramural	1 (6.67%)	—	1 (6.67%)	—	—	2 (2.67%)
HB: Intramural	1 (6.67%)	—	—	1 (6.67%)	—	2 (2.67%)
HB duplicity	—	—	1 (6.67%)	1 (6.67%)	—	2 (2.67%)
SAN: Hemorrhage and infarct-like lesions	—	—	—	—	1 (6.67%)	1 (1.3%)

Age groups: SIUD: before birth; SNUD: 0–30 days; SIDS: 1–12 months; SUDY: 1–35 years; SUDA: >35 years. Abbreviations: M = male; F = female; gw = gestational weeks; *N* = number of cases; hr = hours; dy = days; ms, months; yr, years; SAN = sino-atrial node; HB = His bundle; AVN = atrioventricular node; AVJ = atrioventricular junction; BIF = bifurcation (of HB bundle); CFB = central fibrous body; LBB = left bundle branch; RBB = right bundle branch; — = absent. * Statistically significant: chi-square test; *p* < 0.05.

## Data Availability

Anonymized study data are available on request from the corresponding author. The data are not publicly available due to the privacy and confidentiality in personal data collection and processing, in accordance with EU and international legislation.

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
