# Peer review of "Autopsy for Medical Diagnostics: Finding the Cause of Sudden Unexpected Death through Investigation of the Cardiac Conduction System by Serial Sections"

_diagnostics, 2023, doi:10.3390/diagnostics13111919_

Round 1

Reviewer 1 Report

Overall, the paper is interesting and well written.

However, there are some major points, which are reported below,

The Author state that their results data are useful to understand the cause of death for those unexpected cases that would have otherwise remained unexplained, so to motivate medical examiners and pathologists to perform more in-depth studies. Therefore, they performed an histopathological examination of the cardiac conduction system through serial sections and of the brainstem. However, data reported only refers to Cardiac Conduction System Findings and no data or figures are reported regarding the morphological study of the brainstem (which instead is reported in the method, lines 177-185). authors should include data and histological analysis of the brainstem or alternatively remove this part from the text.

Finally, the study’s methods of investigation of the conduction system are interesting and maybe will have a potential application for any form of sudden death occurring at any age. However, it would be interesting to know whether the authors are aware of other methods/postmortem examinations postmortem imaging investigations that provide a strong complementary tool to the conventional autopsy. This is the case for instance of postmortem cardiac magnetic resonance imaging, which offers a better understanding of the cardiovascular diseases responsible for SCD (Guidi et al. Postmortem Cardiac Magnetic Resonance in Sudden Cardiac Death. Heart Fail. Rev. 2018, 23, 651–665).

Minor points are reported below.

Line 26: “a diagnosis of SUD was assigned to the 75. subjects, which included45 females (60%) and”: remove the point and add a missing space

Line 28: “congenital alterations, of the cardiac”: remove the comma

Please revise the font size in the abstract; it seems that in the first part there is a smaller font the in the second part.

Lines 42-44: “Pallida mors pulsat aequo pede alterno pauperum tabernas regumque turres” _(Pale death knocks with impartial footstep at the cottages of the poor and the palaces of kings) [Horatius QF. Carminum Libri 13 BC; IV: 13–4]. What does this quote refer to?

Line 58: “such as all chemicals that interfere with the way the body’s hormones work, or”; please revise this sentence since it is not clear what does the authors mean. Please add references.

Line 60 and 64: review these sentences to avoid repetitions.

Line 64: 3.5‰; please correct

In the introduction, the authors have made a list of SUD (lines 53-59): SUDY, SIDS, SIUD, SNUD, SUDA. sudden unexpected death of the young (SUDY) and sudden infant death syndrome (SIDS) are repeated twice; please correct. Also, it would be helpful to order these types of SUD in chronological order, such as: intrauterine unexpected death (SIUD), sudden neonatal unexpected death (SNUD), sudden infant death syndrome (SIDS), sudden unexpected death of the young (SUDY), sudden unexpected death in the adult (SUDA).

Why the Author then only describe the SIDS and SIUD? Why sometimes they use the term unexplained, or unexpected, or neither one of these terms in other parts of the paper? Please, try to unify all the terms used (see for instance lines 108-130).

Line 138: 3) SIDS group; please add the number of subjects in this group.

English language and style are fine.

Reviewer 2 Report

The title of this paper is unimaginative. There are many papers with a similar topic. With this approach, it adds almost nothing to the body of knowledge.  It is more like a show-off of how the Lino Rossi Research Center manages cases of sudden, unexpected death. And this is my main comment – what NEW does this study bring?

1.      You should reconsider the definition of SUD – “underlying disease” from the ln 47. Should be explained better. Further, what do you mean by “common” – do you mean “frequent”, or that most deaths are SUD?

2.      Caption to the figures should start off a bit more “formal”  look at the instructioons for authors. 

Round 2

Reviewer 1 Report

The Authors have addressed all the concerns raised by the Reviewer. I have no more academic questions.

Reviewer 2 Report

I think you are good to go.